# Effect of Dry-Aging on Quality and Palatability Attributes and Flavor-Related Metabolites of Pork Loins

**DOI:** 10.3390/foods10102503

**Published:** 2021-10-19

**Authors:** Derico Setyabrata, Anna D. Wagner, Bruce R. Cooper, Yuan H. Brad Kim

**Affiliations:** 1Meat Science and Muscle Biology Laboratory, Department of Animal Science, Purdue University, West Lafayette, IN 47906, USA; dsetyabr@purdue.edu (D.S.); wagne259@purdue.edu (A.D.W.); 2Bindley Bioscience Center, Purdue University, West Lafayette, IN 47907, USA; brcooper@purdue.edu

**Keywords:** dry-aging, loin, pork, metabolomics, consumer sensory, microbial attributes

## Abstract

This study evaluated the effect of dry-aging on quality, palatability, and flavor-related compounds of pork loins. Ten pork loins were obtained at 7 days postmortem, divided into three equal portions, randomly assigned into three different aging methods (wet-aging (W), conventional dry-aging (DA), and UV-light dry-aging (UDA)), and aged for 21 days at 2 °C, 70% RH, and 0.8 m/s airflow. The results showed similar instrumental tenderness values across all treatments (*p* > 0.05), while DA and UDA had a greater water-holding capacity than WA (*p* < 0.05). Both DA and UDA were observed to have comparable color stability to WA up to 5 days of retail display (*p* > 0.05). Greater lipid oxidation was measured in both DA and UDA at the end of display compared to WA (*p* < 0.05). The UV light minimized microorganisms concentration on both surface and lean portions of UDA compared to other treatments (*p* < 0.05). The consumer panel was not able to differentiate any sensory traits and overall likeness between the treatments (*p* > 0.05). Metabolomics analysis, however, identified more flavor-related compounds in dry-aged meat. These findings suggested that dry-aging can be used for pork loins for value-seeking consumers, as it has a potential to generate unique dry-aged flavor in meat with no adverse impacts on meat quality and microbiological attributes.

## 1. Introduction

The pork industry’s focus on growth efficiency has led to the production of leaner and heavier pigs [1]. While improvements in feed efficiency and growth performance have increased the yields and profitability of the swine industry, quality issues in the final products, such as inferior tenderness, juiciness, and flavor, have been reported [2,3,4]. Providing high-quality (palatability) meat products is a vital factor for consumer satisfaction and, in the long term, for the profitability and sustainability of the pork industry [5]. In order to meet consumer expectations for high-quality meat products, post-harvest enhancement techniques, such as brine injection and marination with non-meat ingredients, are often applied to pork products [6,7,8]. Although the application of these techniques has been proven to improve eating quality attributes, there is growing demand for more natural and minimally processed meat products among consumers [9].

Postmortem aging is a natural value-adding process extensively practiced by the meat industry. The application of postmortem aging has been well documented to further improve the sensory attributes of meat, increasing the tenderness, juiciness, and flavor perceived from the products [10]. Across the industry, wet-aging (aging by storing meat in vacuum packaging) is the most commonly utilized aging method. Recently, however, there has been an increasing interest in fresh meat products from dry-aged carcasses or subprimals from value-seeking consumers [11]. Dry-aging is a traditional aging method, where meat is aged without any protective packaging material in a highly controlled environment. In addition to the improvement in both tenderness and juiciness, the application of dry-aging has been reported to generate unique flavors such as “sweet”, “buttery”, and “brown-roasted” in beef, making the final products more desirable [12,13]. The generation of desirable meat flavors has been known to be dictated by the availability of flavor precursors such as amino acids, sugars, nucleotides, and fatty acids in the meat product [14].

The development of high-throughput analysis, such as metabolomics, has enabled comprehensive understanding of biological function through the chemical and biochemical profiling of small compounds (metabolites) in a biological sample. Recently, there has been a growing interest in adopting metabolomics in meat research to gain insights into biochemical and molecular changes of postmortem muscle and their concomitant impacts on meat quality attributes [12,15,16,17]. By utilizing mass spectrometry (MS)-based metabolomics analysis, greater abundance of free amino acids, nucleotides, and sugars were reported in dry-aged beef, potentially explaining the greater flavor observed from dry-aged products [12,18]. Moreover, reduction in off-flavor-related metabolites such as terpenoids and hormones coupled with observed sensory detection were reported, revealing the additional flavor development mechanism following the dry-aging process [18]. These results indicate metabolomics as a novel approach to elucidate and profile flavor-related compounds in meat products.

Currently, dry-aging has been extensively studied in beef products, and only limited research has reported the impacts of dry-aging on pork loin quality attributes [19,20,21,22]. While some levels of conventional chemical analyses along with trained sensory evaluation were conducted, the alteration of flavor precursors and flavor-related metabolites in dry-aged pork loin products have never been profiled. Moreover, given the nature of dry-aging, the presence of microorganisms during the process is inevitable. Consequently, UV lights are often employed by the processors in order to prevent any growth of spoilage bacteria and minimize the microorganism contamination in meat during aging [23,24]. In recent reports, however, it was suggested that the presence of some microorganism could be vital for the development of dry-aging flavor, potentially through the release of proteolytic and lipolytic enzymes into the meat, allowing greater liberation of flavor precursors [25,26]. While UV light application has been shown to be effective in reducing spoilage bacteria and pathogens in various meat applications [27,28], the impact of UV light on dry-aging flavor development is still unclear. Hence, the objectives of this study were to determine the meat quality and consumer acceptance of dry-aged pork loin products and to investigate the flavor precursor differences between dry- and wet-aged pork loins using a novel metabolomics approach.

## 2. Materials and Methods

### 2.1. Sample Collection, Preparation, and Processing

At 7 days postmortem, bone-in and skin-on loins (*M.*
*longissimus thoracis et lumborum,* from 11th–21st vertebrae) were obtained from one side of 10 market-weight pork carcasses (left side, live weight = 117.3 ± 1.7 Kg, crossbreed Landrace x Large White x Duroc, National Pork Board marbling score = 1.4) from Purdue University Meat Laboratory harvest facility. Prior to processing, initial microbiological samples were excised from the lean meat portion of the loin eye (anterior side) of each loin sample, placed in sterile sample bags, and stored in −80 °C until analyses. All loins were then divided into three equal sections (~15 cm) using a band saw and randomly assigned into three aging methods: wet-aging (WA; Clarity Vacuum Pouches Bunzl Processor Division, Riverside, MO, USA), conventional dry-aging (DA), and UV-light dry-aging (UDA).

All sections were measured for initial pH and weight prior to 21 days of aging at 2 °C, 70% relative humidity, and 0.8 m/s airflow. The samples were placed on food-safe racks (Uline, Pleasant Praire, WI, USA) for the aging process. The UDA samples were exposed to UV-light treatment twice each day with a dose of 5 J/cm^2^ per treatment. The UV lights (Phillip TUV T8 UVC light, Eindhoven, Netherlands) were mounted 30 cm above the loins and turned on for 5 min per treatment. Sections were rotated weekly to reduce location variation during aging. At the end of aging, sections were measured for final weight. All sections were then skinned, deboned, trimmed of any dehydrated surface, and weighed for the final yield estimation. Microbial samples were collected from trimmed dehydrated surfaces and lean portion of each loin by immediately excising the inner lean meat portion following trimming. The samples were placed in sterile sample bags and stored in −80 °C until analyses, similar to the initial microbiological samples. The trimmed loins were then measured for their pH and cut into multiple chops for further meat quality (2.54 cm thick) and biochemical analyses (1.27 cm thick). Except for the chops assigned for the color display and drip loss analyses, all chops were vacuum-packed individually and stored in a −80 °C freezer until analyses.

### 2.2. Aging Loss, Processing Loss, and Saleable Yield

The aging loss was measured by calculating the weight differences before and after the aging treatments to observe the shrink/water loss during aging. Final weights were collected for the sections to calculate trimming loss and final yield following the trimming process. All of the losses were presented as percentage loss.

### 2.3. pH Measurement

The pH was measured using a hand-held meat pH meter (HANNA HI 99163, Hanna Instrument, Inc., Warner, NH, USA) before and after the aging treatments. The probe was inserted directly into the meat in two different locations. The pH meter was calibrated according to the manufacturer’s guidelines before any measurement.

### 2.4. Water-Holding Capacity Measurement

The water-holding capacity (WHC) was measured by measuring drip loss, display loss, freeze/thaw loss, and cook loss. All measurements were expressed as percent loss, measuring the weight changes between the initial and final weight of the samples following each procedure. All samples were blotted dry using paper towels prior to any weight measurement.

The drip loss was measured using the Honikel method [29] with the modification described by Kim et al. [30]. In brief, 40 g of meat was collected from each sample. The cubes (about 2.54 × 2.54 × 2.54 cm) were trimmed of any visible connective tissues and fat and were then suspended using netting for 48 h in airtight containers at 2 °C. Immediately after, the final weights of the samples were measured to calculate the drip loss (%).

The display loss was measured on the chops designated for color display simulation. Chops were weighed before display and were then re-weighed following the 7 days color display.

For freeze/thaw loss, samples designated for cook loss and Warner–Bratzler shear force (WBSF) were utilized. The frozen samples were thawed at 2 °C for 24 h prior to final weight measurement. The loss was determined by calculating the differences between the weight before and after the freezing and thawing process.

The cook loss was observed by cooking the sample to an internal temperature of 71 °C using a clamshell grill (dual-sided grill) (Griddler GR-150, Cuisinart, Glendale, AZ, USA) and monitored using a T type thermocouple (Omega Engineering, Stamford, CT, USA) connected to an OctTemp 2000 data logger (Madge Tech, Inc., Warner, NH, USA). When the internal temperature was reached, samples were removed from the griddle and rested for 10 min prior to weighing for the final weight. Samples were then wrapped using aluminum foil and kept in a 4 °C refrigerator overnight for WBSF measurement.

### 2.5. Warner–Bratzler Shear Force Measurement

The shear force was measured by collecting a total of 8 cores parallel to the muscle fiber direction from each chop. The cores (1.27 cm diameter) were cut perpendicular to the muscle fiber using TA-XT Plus Texture Analyzer (Stable Micro System Ltd., Godalming, UK) using the V-shaped blade attachment for the WBSF measurement. The crosshead speed was set to 3 mm/second, and a 2.5 Kg load cell was utilized during the measurement. The average peak shear force (*N*) from the cores was calculated.

### 2.6. Display Color Stability

One chop from each section was collected for simulated color display. The chops were placed on Styrofoam trays with drip pad, overwrapped using PVC film (Reynolds Food Service Packaging, Richmond, VA, USA), and displayed for 7 days under light (1800 lx, color temperature = 3500 K, OCTRON^®^ T8 Lamps, Osram Sylvania LTD., Markham, ON, Canada) at 2 °C. The samples were evaluated daily for color using Hunter MiniScan EZ colorimeter (Hunter, Reston, VA, USA), measuring the CIE L*, a*, and b* on three random locations on the surface of the chops. The instrument was calibrated following the manufacturer’s guidelines and equipped with a 25 mm (diameter) port opening prior to any data collection. Illuminant A was used, and the observer was set to standard 10°. Hue angle and Chroma value were calculated using the following formulas: hue angle = tan − 1(b*/a*) and Chroma = (a*2 + b*2)½ [31].

### 2.7. Lipid Oxidation

Prior to the analysis, the whole fresh pork chops assigned for biochemical analysis were minced, submerged into liquid nitrogen, and pulverized using a blender (Waring Products, CT, USA). The lipid oxidation extent of the samples was measured through the 2-thiobarbituric reactive substances (TBARS) assay described by Buege and Aust [32] with modification by Setyabrata and Kim [33]. In brief, 5 g of the pulverized samples was homogenized in 15 mL of distilled water and 50 µL of 10% butylated hydroxyl anisole. Following homogenization, 1 mL of the homogenate was added to 2 mL of 20 mM 2-thiobarbituric acid solution in 15% tricholoroacetic acid solution. The samples were then mixed and heated in an 80 °C water bath for 15 min. The samples were removed and cooled in ice water for 10 min prior to centrifugation at 2000× *g* for 10 min. After centrifugation, the supernatant was filtered through a Whatman Filter Paper No. 4 (Cytiva, Marlborough, MA, USA). The samples’ absorbance was then read at 531 nm using Epoch™ Microplate Spectrophotometer (BioTek Instrument Inc., Winooski, VT, USA). The TBARS value was calculated using a molecular extinction coefficient (1.56 × 105 M^−1^ cm^−1^) and expressed as mg malondialdehyde/kg meat. Lipid oxidation was measured on samples collected before and after the display.

### 2.8. Microbial Analysis

The microbial analyses were conducted following the method described by Setyabrata et al. [34]. The microbial analysis was performed on the initial (prior to aging), surface, and lean portions from each sample collected after aging treatments. The samples were thawed for 6 h prior to the analyses. In brief, 5 g of sample was aseptically collected and placed into a stomacher bag (WhirlPak, Madison, WI, USA) with 50 mL 0.1% peptone water (BD Difco™, Sparks, MD, USA). The samples were then hand stomached for 1 min. The rinsate was collected and serially diluted using a 1:10 dilution factor with the dilution range of 10^0^ to 10^−5^. All dilutions were then plated in duplicate into plate count agar (BD Difco™, Sparks, MD, USA) for total aerobic bacteria plate count (APC); de Man, Rogosa, and Sharpe agar (BD Difco™, Sparks, MD, USA) for lactic acid bacteria (LAB); and Yeast and Mold films (Petrifilm™, 3M Microbiology Products, St. Paul, MN, USA) for both yeast and mold enumeration. After inoculation, the APC plates were incubated at 37 °C for 48 h. The LAB plates were incubated under anaerobic conditions generated using anaerobic packs (Oxoid™ AnaeroGen, Waltham, MA, USA) for 72 h at 37 °C. The yeast and mold films were incubated at 25 °C for 120 h. After the designated incubation period, colonies were counted, and the final concentration was expressed as log10 CFU/mL of rinsate. For APC and LAB measurement, plates with colonies count below 25 colonies on the lowest dilution were considered to have bacterial concentration below detection limit (BDL). For the yeast and mold petrifilm, the detection limit was set at 15 colonies per the manufacture’s recommendation.

### 2.9. Consumer Sensory Analysis

The consumer sensory evaluation was conducted at Purdue University, and the exemption was approved by Purdue University Institutional Review Board (#IRB-2019-16). The consumer sensory evaluation was conducted using 120 panelists recruited from the community surrounding the West Lafayette, Indiana area.

The samples collected for the sensory analysis were thawed at 2 °C overnight before the sensory session. All samples were cooked using a clamshell grill (Griddler GR-150, Cuisinart, Glendale, AZ, USA) until the internal temperature reached 71 °C. Following the cooking process, chops were trimmed from any visible fat and connective tissues. The chops were then cut into 1 cm × 1 cm × 2.54 cm cubes, placed into a sample cup with a lid and kept in a warmer held at 60 °C for no longer than 15 min prior to serving. The samples were served under a red incandescent light, and panelists were supplied with water and unsalted saltine crackers as a palate cleanser. A starter chop (wet-aged, 2 weeks) was also served and evaluated first prior to performing the actual samples to help the panelist adjust to the evaluation process.

Prior to the sample evaluation, a basic demographic survey was conducted. For the sensory evaluation, samples were scored using an unstructured hedonic test with a scale of 0 to 100 points (0 as dislike extremely, 50 as neither like or dislike, and 100 as like extremely) to observe the flavor, tenderness, juiciness, and overall liking. Additionally, the panelists were also asked to rate the acceptability (acceptable or unacceptable) of each attribute tested and each sample’s perceived quality (unsatisfactory quality, everyday quality, better than everyday quality, and premium quality). After all samples were evaluated, a questionnaire regarding the dry-aging process, pork dry-aging, and willingness to pay was provided as an end survey.

### 2.10. Metabolomics Analysis

#### 2.10.1. Metabolite Extraction

A total of 5 samples were randomly selected from each treatment for the metabolomics analysis. The metabolomics analysis was conducted using the previously homogenized biochemical samples, as described in Section 2.7. Briefly, 200 mg of homogenized sample was mixed with 300 µL of methanol in a tube containing ceramic beads. The samples were then extracted using a Precellys 24 tissue homogenizer (Bertin Instruments, Bretonneux, France). A total of 3 cycles was used to extract the sample, each running for 30 s at 6500 rpm with 30 s rest in between the cycles. Following the extraction, 300 µL of chloroform was added to the tube and mixed for 10 s. Water (300 µL) was then added, and the tubes were placed on a shaker for 15 min at 4 °C. The tubes were then centrifuged at 1000× *g* for 5 min to separate the layers. The upper layer was then collected, transferred to a vial, and dried using a SpeedVac Concentrator (Thermo Scientific, Waltham, MA, USA).

#### 2.10.2. Ultra-Performance Liquid Chromatography-Mass Spectrometer Analysis

Prior to chromatographic separation, the dried samples were reconstituted in an aqueous solution (95% water with 5% acetonitrile) containing 0.1% formic acid. The samples were then separated using similar conditions described by Setyabrata et al. [18]. The reconstituted samples were separated using an Agilent 1290 Infinity II UPLC system (Agilent Technologies, Palo Alto, CA, USA) equipped with a Waters Acquity HSS T3 (2.1 × 100 mm × 1.8 um) separation column (Waters, Milford, MA, USA) and an HSS T3 (2.1 × 5 mm × 1.8 um) guard column. The sample injection volume was set to 5 μL. The binary mobile phase consisted of solvent A (0.1% formic acid (*v*/*v*) in ddH2O) and solvent B (0.1% formic acid (*v*/*v*) in acetonitrile). The column was maintained at 40 °C with the mobile phase flow kept at 0.45 mL/minute. Initial conditions of 100:0 A:B were held for 1 min, followed by a linear gradient to 70:30 over 15 min, followed by a linear gradient to 5:95 over 5 min, with a 5:95 hold for 1.5 min. Column re-equilibration was performed by returning to initial starting conditions of 100:0 over 1 min, with a hold for 5 min. Following the separation, the sample was identified using Agilent 6545 quadrupole time-of-flight (Q-TOF) mass spectrometer (Agilent Technologies, Santa Clara, CA, USA), with positive electrospray ionization (ESI) mode applied for mass spectral (70–1000 m/z) data collection. The collected data were analyzed using Agilent MassHunter B.06 software (Agilent Technologies, Santa Clara, CA, USA), and the mass accuracy was improved by infusing Agilent Reference Mass Correction Solution (G1969-85001; Agilent Technologies, Santa Clara, CA, USA). Peak deconvolution was performed using Agilent ProFinder (v B.08). Peak identification was improved by applying data-dependent MS/MS collection on composite samples with 10 eV, 20 eV, and 40 eV collision energy. The metabolites were identified by comparing them to the human metabolome database (HMDB; www.hmdb.ca, accessed on 1 September 2021), with a tolerance of 0.1 Da for MS1 and 0.5 Da for MS2.

### 2.11. Statistical Analysis

The experimental design of the current study was a randomized complete block design. The animal was considered as the random effect, and the different aging treatments were considered as the fixed effect in the model. Sample location source was added as a fixed effect during the microbiological analysis to identify potential location effect. The period effect was also added as a fixed effect for the color and oxidative analyses. Both panelists and sessions were added as a random effect for the sensory evaluation. The data were analyzed using PROC GLIMMIX procedure from SAS 9.4 software (SAS Institute Inc., Cary, NC). The least-square means were separated, and the significance level was defined at the level of *p* < 0.05.

Metabolomics analysis were conducted using MetaboAnalyst 5.0 [35]. Metabolite features with missing values were given a small value using half of the minimum value in the original samples. The data were then normalized by log transformation and were scaled using the auto-scaled option (mean-centered and divided by the standard deviation). The metabolomics data were then subjected to ANOVA with Tukey post hoc testing, principal component analysis (PCA), and hierarchical cluster analysis (HCA) with ward clustering methods.

## 3. Results and Discussion

### 3.1. Processing Loss and Total Yield

Greater processing loss was observed from both DA and UDA loins compared to WA loins (*p* < 0.05, Table 1), leading to a higher total yield for WA treatments compared to the other treatments (*p* < 0.05). The processing loss consisted of shrink/purge loss, crust loss, fat/skin loss, and bone loss. No differences were observed for fat/skin loss and bone loss for all the treatments (*p* > 0.05). Both DA and UDA samples, however, had more shrink/purge loss compared to WA samples (*p* < 0.05). Consequently, more crust loss was also observed from both DA and UDA loins compared to WA loins (*p* < 0.05).

A decrease in product yields from dry-aged treatments has been constantly reported in previous studies (though predominantly dry-aged beef) [12,20,36]. The substantial decrease in yield is expected, mainly due to the moisture evaporation during the dry-aging process and the removal of the dehydrated surfaces (crust) following the aging process. Our current results were in agreement with Berger et al. [36], where the authors reported no significant treatment differences in both bone and fat losses for grass-fed beef loins aged using WA, DA, and dry-aging in water-permeable bag methods. In the current study, the skin was kept intact for all treatments during the aging process. It was previously suggested that adding a barrier, such as a moisture-permeable bag, helped minimize the moisture loss during aging [37,38]. For pork dry-aging, thus, it could be surmised that the presence of the skin during the aging process could act as an additional barrier to limit moisture loss and environmental exposure. Further study to identify the functionality of the skin during dry-aging would be beneficial to increase profit and yield from the process.

### 3.2. pH, Water-Holding Capacity, and Shear Force

Higher pH (*p* < 0.05, Table 2) was measured in UDA (5.62) compared to WA and DA (5.58 and 5.59, respectively) samples. Currently, there is still inconsistency in the literature in regards to pH changes following dry-aging treatment in pork. A previous study by Hwang et al. [39] reported an increase in pH following dry-aging process when compared to wet-aging, showing pH values of 5.91 and 5.73, respectively. Similarly, J. H. Kim et al. [20] also reported a higher pH value in DA pork compared to WA pork, showing pH values of 5.51 and 5.41, respectively. On the other hand, Jin and Yim [22] found the pH was not affected by dry-aging application in pork compared to WA, showing pH values of 5.70 and 5.74, respectively. It was suggested that the presence of microbes during the process contributed to the change of pH in the product, whether through the generation of acid or the release of nitrogen products [26,40,41]. While the pH was significantly affected in this study, the changes observed were numerically minimal (<0.05 unit difference), and thus its impacts on meat quality attributes would be practically less meaningful.

For water-holding capacity (WHC), it was found that WA chops had reduced WHC compared to both DA and UDA chops, indicated by higher drip loss, display loss, and freeze/thaw loss measured in WA chops (*p* < 0.05; Table 2). Based on the current results, it could be postulated that the greater moisture loss during the dry-aging process decreased the available moisture in the product, hence limiting the meat water loss in subsequent processes. No difference was found for cook loss among all the treatments (*p* > 0.05).

The different aging treatments did not affect the WBSF of the samples (*p* > 0.05, Table 2), having comparable shear force values of 26.41, 25.08, and 27.05 N for WA, DA, and UDA loins, respectively. Similar results were reported by Hwang and Hong [21], where the shear force values of unpressurized DA pork loins aged for 21 days were not different from those of its WA counterpart, and by Juárez et al. [19], where the shear force of DA and WA pork loins were not different following 14 days of aging. Likewise, Kim et al. [12], Berger et al. [36], and Dikeman et al. [42] also reported that DA beef exhibited no difference in shear force and texture profile compared to its WA counterpart. This indicated that different aging treatments would not affect the extent of proteolysis in the product, while the length of the aging period might be more influential on product tenderness.

### 3.3. Display Color Stability

A significant treatment and period interaction was observed on all instrumental color traits measured, except for b* (yellowness) and Chroma, during the 7-day retail display (Figure 1). The lightness (L*) was found to be generally lower (*p* < 0.05) for both DA and UDA chops throughout the retail display compared to WA chops. The redness (a*) was initially comparable (*p* > 0.05); however, significantly lower redness was then identified in both DA and UDA chops on days 4, 5, and 7 of the display. More discoloration was detected in both DA and UDA samples from day 5 until the end of the display compared to WA samples (*p* < 0.05), indicated by the hue angle values.

The lower lightness in dry-aged products could be associated with the greater moisture loss in the product. Lower moisture has been associated with less surface moisture availability, leading to a decrease in light reflection and thus resulting in a darker appearance [12]. These findings are in agreement with previous pork dry-aging studies, where higher initial lightness in wet-aged products compared to dry-aged products was reported [20,22,39]. While changes in color and color stability could be detrimental, studies have reported that pork consumers preferred darker-colored chops over lighter-color chops [43,44,45], suggesting potential dry-aging benefits during retail settings. Although there has been consistency in terms of meat lightness following the dry-aging process, conflicting results were reported for a* value. Previous studies by Jin and Yim, and Hwang et al. [22,39] reported that dry-aged pork had a higher a* value compared to the wet-aged counterpart. On the other hand, similar to the current study, Kim et al. [20] reported a decrease in a* value of dry-aged pork compared to wet-aged pork. This discrepancy could potentially be attributed to the pH of the meat samples utilized between the studies. Both Jin & Yim and Hwang et al. [22,39] reported a higher pH value (~5.9) compared to the value measured in the current study and the study by Kim et al. [20] (~5.6). Meat with lower ultimate pH is often observed to have inferior color stability when compared to high–ultimate pH meat [46], mainly due to a decrease in the redox stability of myoglobin [45].

In terms of the color stability of dry-aged meat, only limited information is currently available in the literature. Previous studies in beef reported a significant increase in sensory discoloration [34,47] and instrumental discoloration (Hue angle) [34] in dry-aged beef compared to wet-aged beef during simulated retail display. Similarly, an increase in instrumental discoloration in dry-aged treatments was also identified in the current study, starting from day 5 of the display until the end of the display.

### 3.4. Lipid Oxidation

A significant period and aging treatment interaction was identified for lipid oxidation of the samples based on the TBARS analysis (Figure 2). The lipid oxidation was increased over the simulated display regardless of the treatments (*p* < 0.05). Prior to the display, no difference was observed across all treatments (*p* > 0.05). Following the display, however, greater lipid oxidation (*p* < 0.05) in DA and UDA samples was measured when compared to WA samples.

The results of the current study indicated that dry-aging altered the oxidative stability of the product. Although no immediate effect was observed, dry-aged products were more susceptible to oxidation, demonstrated by the greater extent of lipid oxidation following the retail display period. Corroborating the current TBARS observation, greater discoloration was also observed on both DA and UDA chops following the color display, indicating a general loss of reducing capability in the meat. It is possible that during the dry-aging process, the environmental exposure initiated the oxidation process in the meat and began the accumulation of radical oxygen species (ROS). While limited, the presence of the ROS could then accelerate further oxidation [48], decreasing the oxidative stability following the dry-aging process.

Interestingly, no difference was observed in the oxidative stability between DA and UDA loin samples. It was previously suggested that UV light application would further induce the extent of oxidation through photo-oxidation [49]. Similar results were previously reported by Setyabrata et al. [34] for dry-aged beef lipid oxidation. Those authors found that the UDA treatment had a similar color to both WA and DA treatments both before and after 7 days of aerobic display storage. It was suggested that the presence of the dehydrated surface acted as a barrier to limit the extent of oxygen transfer and light penetration, therefore minimizing the oxidative impact of UV light.

### 3.5. Microbial Analysis

The initial samples were found to have microbial concentration below the detection limit for all microbial groups measured (data not shown), indicating a microbial load in the product comparable to prior to the aging process. A significant aging treatment and location interaction, however, was observed following the aging across all of the treatments (Table 3).

The WA surface was found to have the greatest concentration of APC (2.69 log10 CFU/mL, *p* < 0.05), followed by DA crust (1.37 log10 CFU/mL), while UDA crust and lean portions had a similar APC concentration (*p* > 0.05). WA surface was also identified to have the highest LAB concentration (2.33 log10 CFU/mL) compared to all other treatments (*p* < 0.05). UDA crust and DA lean had concentrations below detection limit for LAB. Both WA and DA crust had no significant differences for mold concentration (1.82 log10 CFU/mL and 1.39 log10 CFU/mL, respectively); however, both were higher compared to other treatments (*p* < 0.05). Yeast was only detected on the surface crust of WA and DA treatments, and there was found to be no difference between the two treatments (*p* > 0.05).

Generally, greater microbial concentration was measured in the crust portion of the sample and was reduced following the trimming process. Within the crust portion, UDA had the lowest microbial concentration compared to the other treatments, indicating that UV light suppressed microbial growth. Following trimming, however, minimal microbial concentration was detected in the lean portion of the UDA group. Similar results were previously presented by Li et al. [38], where a higher concentration of both APC and LAB count was observed in the inner portion when compared to the surface portion after the trimming process. No explanation, however, was provided by those authors. While it is still unclear, it has been suggested that the attached microbes could penetrate into the meat utilizing the gaps between muscle fibers generated during the aging process, thus contaminating the inner portion of the meat [50,51,52]. Additionally, microbe-induced proteolysis was suggested to increase the extent of penetration and allow more microbes to migrate into the inner portion [52]. Furthermore, UV light can only affect the areas exposed to the light, and thus still allows microbial growth in unexposed areas (e.g., within meat fold, knife cuts).

### 3.6. Demographic and Survey Data

The consumer demographic data is available in Appendix A. The panelists were mainly between 20–29 years old (70%). The consumer panelists responded that they mainly consumed pork 1–5 times/week (87.5%). A total of 45.8% of the panelists considered flavor the most important palatability trait when consuming pork products, while both juiciness and tenderness shared a similar percentage (26.7% and 27.5%, respectively). In addition, the panelists in this study indicated that they preferred pork cooked to medium doneness (31.7%), followed by a split between medium-well (25.8%) and well done (26.7%) degree of doneness. Following the sample evaluation, panelists were presented with survey questions involving dry-aging and their willingness to pay for dry-aged pork (Table 4). From all of the panelists, 52.5% had previously tried and consumed dry-aged products. It was indicated that restaurant was the primary method to obtain the dry-aged products (39.7%), followed by the local supermarket (23.8%). Most of the panelists agree that meat aging is a positive term (85.8%), and the dry-aging process itself is perceived to generate products with similar safety with other meat products (65.8%). When asked about the willingness to pay for the dry-aged pork product, 55.8% of the consumer panel were willing to pay $1.00 more per 1 lb. (0.45 Kg) of dry-aged pork.

It is of interest to note that the majority of the panelists rated flavor as the most important palatability attribute when consuming pork products compared to tenderness and juiciness. Both juiciness and tenderness are often considered as the main palatability attributes critical for pork acceptability, motivating researchers to work on minimizing these sensory issues [53,54,55]. While there are studies focusing on pork flavor, most of those studies focused on the reduction off-flavor development and were not yet looking at the different precursors of desirable pork flavors [5,56,57,58,59]. The current results indicate that there might be a potential shift in consumer preferences in pork palatability as more improvements are observed in both juiciness and tenderness attributes of pork products. Supporting the current observation, a previous consumer perception survey in Italy also reported higher preferences of potential purchase for dry-aged pork loins, further indicating the change in interest among pork consumers [60]. Additionally, similar shifts showing an increased focus in flavor by the consumers have also been repeatedly reported in beef products [61,62,63,64], demonstrating a general increase in flavor interest by the consumers.

### 3.7. Consumer Panel Evaluation

The consumer panelists found that the different aging methods generated products with comparable sensory attributes (*p* > 0.05, Table 5). Similar scores were given by the panelists for flavor, tenderness, juiciness, and overall liking, regardless of the treatments. The products were also rated to have similar acceptability in all the traits tested (*p* > 0.05), with all considered to have acceptable tenderness, juiciness, flavor, and overall acceptability. For the perceived quality, the majority of the samples were considered as everyday quality by the consumer panelists. More panelists, however, rated WA to have premium quality compared to DA (*p* < 0.05), while UDA was not different from both WA and DA (*p* > 0.05).

While sensory evaluation is routinely reported for dry-aged beef, only limited information regarding dry-aged pork is available in the literature. A previous study by Lee et al. [65] reported that experienced panelists scored 40-days-dry-aged pork, which had higher taste, flavor, texture, and overall acceptability scores when compared to unaged pork products. Similarly, Kim et al. [20] also reported that the trained panel found greater aroma, higher juiciness, and lower off-flavor in dry-aged pork compared to the wet-aged counterpart aged to both 7 and 14 days. Although positive dry-aged sensory improvements were observed by trained panel evaluation in previous studies, the current consumer panel results do not show any significant differences for all sensory traits between dry-aged and wet-aged pork. This observation could potentially be attributed to the unfamiliarity of consumers with the dry-aged pork taste [60]. To our knowledge, this is the first study to report consumer likeness of dry-aged pork products. Additional research, including trained panel evaluation to profile descriptive sensory attributes of dry-aged pork, would be necessary to determine the impacts of dry-aging on specific organoleptic properties of pork loins.

### 3.8. Metabolomics Analysis

A total of 1839 metabolite features were observed via the untargeted UPLC-MS analysis. Following the statistical analysis, 197 metabolites were found to be significantly influenced by the aging treatments (*p* < 0.05, FDR < 0.05) and were then utilized as a subset for further analysis. Principle component analysis (PCA) of the metabolites revealed a clear clustering of all treatments (Figure 3a). A distinct separation between dry-aging treatments and wet-aging was exhibited along the PC1 axis, explaining 65.9% of the variation observed between the aging types. Further separation within the dry-aging treatments was observed across the PC2 axis, explaining 11.5% of the variation and demonstrating metabolite profile difference between the DA and UDA treatments. Likewise, HCA (Figure 3b) presented a more comparable metabolite profile between DA and UDA loins when compared to WA loins.

Of the 197 metabolites, 27 features were identified through MS/MS spectral matching with the HMDB database. A total of 13 metabolites were found to be significantly more abundant in either DA or UDA treatments, 10 metabolites were greatly abundant in the WA treatment, and 4 metabolites were found to be abundant in both WA and DA treatments. Those metabolites could be loosely separated into 4 different groups and presented in Table 6. More protein/amino-acid-derived metabolites were found in the dry-aged treatments compared to the WA treatment, including histidine, nitrotyrosine, methylcrotonylglycine, and phenylalanine. Likewise, more nucleotide-derived metabolites (dihydrothymine, thymidine, cyclic AMP, IMP, hypoxanthine, and cytidine) were identified and observed to be in higher abundance for both DA and UDA samples compared to the WA samples. Interestingly, greater concentrations of antioxidant compounds (hydroquinone, niacinamide, and pelargonidin) were observed in WA and DA samples compared to UDA samples.

Greater abundance of amino acids and nucleotides have been suggested to positively influence meat flavor, mainly by acting as flavor precursors involved in Maillard reaction during the cooking process [66,67]. However, in the current study, the consumer panel did not find any differences in sensory traits between different aging methods. It might be due to the fact that although dry-aged meat contained a higher abundance of flavor precursors, the flavor volatiles might not be adequately generated during the cooking process to influence flavor perception. Previous studies found that the volatile generation from Maillard reaction is not only dependent on the concentration of the substrates but also on the environmental condition such as pH, water activity, and temperature [68,69]. Reports had also indicated that consumers rated higher for the flavor attribute when the pork product had a higher pH (>5.8) and was cooked to a lower degree of doneness [54,55], providing flavor descriptors such as sweet and less acidic [70], potentially due to greater volatile generation in the product. The samples in the current study, however, were cooked to an internal temperature of 71 °C (medium doneness) and were observed to have a pH range of around ~5.6. It was suggested that lower pH increases the presence of protonated amino groups, decreasing the reactivity during Maillard reaction and therefore influencing the final volatile concentration [71]. Additionally, the lower pH condition was also reported to decrease the presence of pyrazines, thiazoles, and furans volatiles, which have been known to contribute to the meaty and roasted flavors [14,69,72,73] often associated with dry-aged products. Similarly, changes in the water activity and cooking temperature have also been suggested to alter the rate and type of Maillard reaction in the product [71,74], impeding the volatile production during the cooking process. Subsequent studies to expand the effect of different meat conditions on dry-aging flavor production would be crucial to understand further the mechanism involved in the flavor production.

Other than flavor precursors, hydroquinone, niacinamide, and pelargonidin were present in a greater abundance in both WA and DA samples compared to UDA. These metabolites have been previously identified to display antioxidant capability [75,76,77]. The loss of antioxidant availability in UDA could potentially be attributed to the application of UV light during the dry-aging process. The reduction of antioxidant compounds in UDA treatment was expected as the samples were exposed to UV light, which is known to induce oxidation through photo-oxidation [49].

Shikimic acid was also identified in the samples through metabolomics analysis. This metabolite was previously reported to act as an intermediate compound involved in the biosynthesis of aromatic amino acids (l-phenylalanine, l-tryptophan, and l-tyrosine) by microorganisms through the shikimate pathway [78]. Interestingly, this pathway is only observed in plants and microorganisms and is not observed to be present in animal metabolism. Currently, the role of microorganisms during the dry-aging process is still obscure. Microorganisms have been well known to release proteolytic and lipolytic enzymes to further promote muscle degradation. The observation of this compound, however, suggested that microorganisms could also participate in flavor development by directly producing the flavor pre-cursor and are not limited to muscle degradation activity.

## 4. Conclusions

In the current study, dry-aging of fresh pork loins resulted in similar instrumental tenderness, greater WHC, and lower microbial concentrations compared to conventional WA samples. The application of UV lights during dry-aging was also identified to further minimize the presence of microorganisms with minimal impact on meat quality. Untargeted UPLC-MS metabolomics analysis determined that a greater abundance of flavor-related precursors (amino acids and nucleotides) were liberated in both dry-aging treatments compared to conventional WA products. While this result could suggest potential development of unique dry-aged flavor in the dry-aged pork loins, the consumer panel was not able to find sensory trait differences across all aging treatments. Hence, additional studies utilizing a trained (focus group) panel to conduct descriptive sensory analysis along with other volatile chemical analysis would be of interest to further elucidate the dry-aging flavor volatile generation and their impact on the dry-aged pork’s organoleptic properties.

## Figures and Tables

**Figure 1 foods-10-02503-f001:**
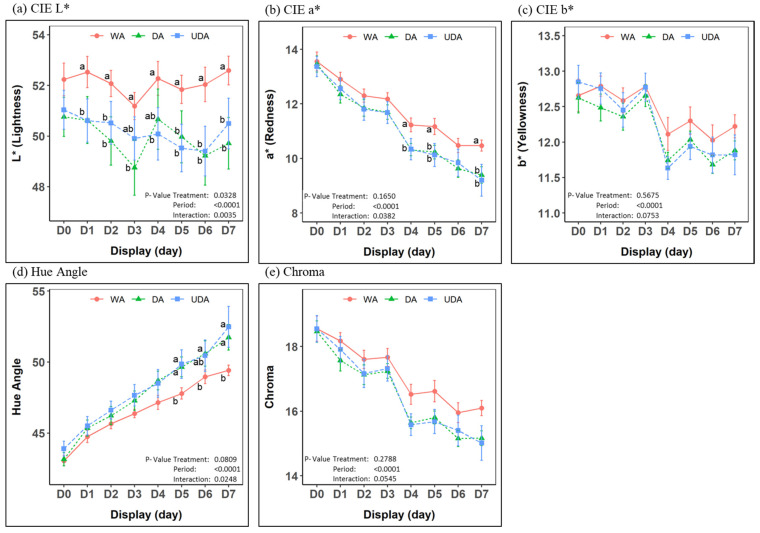
Effect of different aging treatments on the instrumental color characteristic of pork loins (*M*. *Longissimus lumborum*) aged for 21 days during 7 days of display period. Different aging treatments: wet-aging (WA), conventional dry-aging (DA), and UV-light dry-aging (UDA). ^a,b^ Means with different letters indicate significant differences within the same display day (*p* < 0.05). (**a**) CIE L*; (**b**) CIE a*; (**c**) CIE b*; (**d**) Hue Angle; (**e**) Chroma.

**Figure 2 foods-10-02503-f002:**
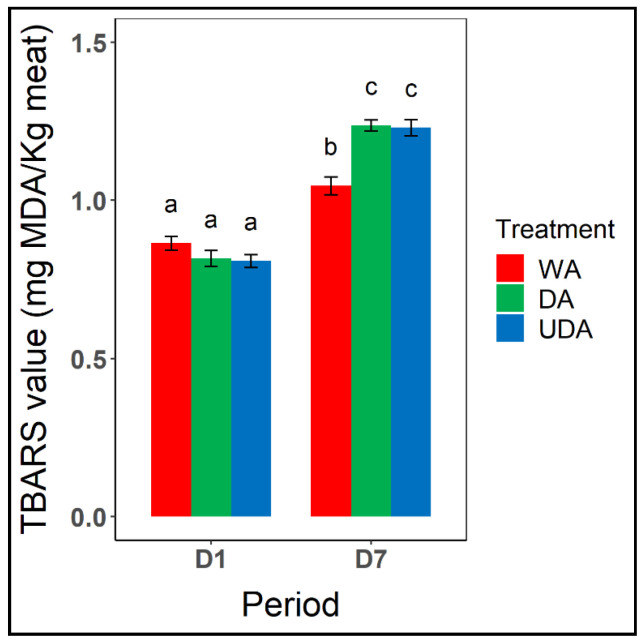
Effect of different aging treatments on lipid oxidation of pork loins (*M*. *Longissimus lumborum*) aged for 21 days. Different aging treatments: wet-aging (WA), conventional dry-aging (DA), and UV-light dry-aging (UDA). ^a–c^ Means with different letters indicates significant differences within the same display day (*p* < 0.05).

**Figure 3 foods-10-02503-f003:**
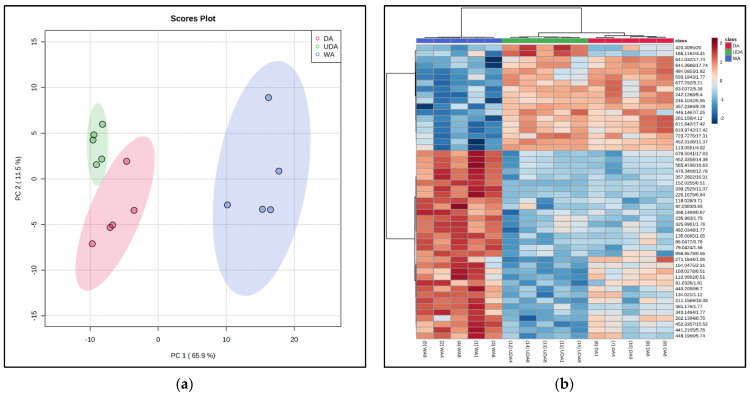
Principle component analysis (PCA, (**a**)) and hierarchical clustering analysis (HCA, (**b**)) of significant metabolites from pork loins (*M*. *Longissimus lumborum*) aged for 21 days with different aging treatments: wet-aging (WA), conventional dry-aging (DA), and UV-light dry-aging (UDA).

**Table 1 foods-10-02503-t001:** Effect of different aging treatments on shrink/purge loss, crust loss, fat/skin loss, bone loss, total loss, and total saleable yield of pork loins (*M. Longissimus lumborum*) aged for 21 days.

Treatments	Shrink/Purge Loss (%)	Crust Loss (%)	Fat/Skin Loss (%)	Bone Loss (%)	Total Processing Loss (%)	Total Yield (%)
WA	3.20 ^b^	0.00 ^b^	34.57	21.02	45.04 ^b^	54.96 ^a^
DA	16.13 ^a^	8.52 ^a^	34.23	23.50	58.54 ^a^	41.46 ^b^
UDA	16.47 ^a^	8.29 ^a^	32.35	22.21	59.12 ^a^	40.88 ^b^
SEM	0.65	0.53	2.18	1.52	1.78	1.78
*p*-value	<0.0001	<0.0001	0.7439	0.1996	<0.0001	<0.0001

^a,b^ Different superscript letters indicated a significant difference between the different aging methods (*p* < 0.05). Different aging treatments: wet-aging (WA), conventional dry-aging (DA) and UV-light dry-aging (UDA). SEM: standard error of means.

**Table 2 foods-10-02503-t002:** Effect of different aging treatments on pH, water-holding capacity measurements, and shear force of pork loins (*M*. *Longissimus lumborum*) aged for 21 days.

Treatments	pH	Drip Loss (%)	Display Loss (%)	Freeze/Thaw Loss (%)	Cook Loss (%)	Shear Force (N)
WA	5.58 ^b^	1.42 ^a^	4.37 ^a^	2.86 ^a^	21.92	26.41
DA	5.59 ^b^	0.85 ^b^	3.57 ^b^	1.92 ^b^	22.83	25.08
UDA	5.62 ^a^	0.77 ^b^	3.48 ^b^	1.79 ^b^	21.98	27.05
SEM	0.0117	0.1611	0.2562	0.2426	0.7548	1.3044
*p*-value	0.0311	0.0159	0.0285	0.0083	0.5573	0.3274

^a,b^ Different superscript letters indicated a significant difference between the different aging methods (*p* < 0.05). Different aging treatments: wet-aging (WA), conventional dry-aging (DA) and UV-light dry-aging (UDA). SEM: standard error of means.

**Table 3 foods-10-02503-t003:** Effect of different aging treatments on total aerobic bacteria (APC), lactic acid bacteria (LAB), and mold and yeast concentration on the crust (surface) and the lean portion of pork loins (*M. Longissimus lumborum*) aged for 21 days.

Location	Treatment	APC (log_10_ CFU/mL Rinsate)	LAB (log_10_ CFU/mL Rinsate)	Mold (log_10_ CFU/mL Rinsate)	Yeast (log_10_ CFU/mL Rinsate)
Lean	WA	0.72 ^c^	0.40 ^b^	BDL	BDL
	DA	0.54 ^c^	^1^ BDL	0.13 ^b^	BDL
	UDA	0.29 ^c^	0.14 ^b^	0.17 ^b^	BDL
Surface/Crust	WA	2.69 ^a^	2.33 ^a^	1.82 ^a^	0.24
	DA	1.37 ^b^	0.10 ^b^	1.39 ^a^	0.64
	UDA	0.15 ^c^	BDL	BDL	BDL
SEM		0.25	0.13	0.26	0.19
*p*-value	Treatment	0.0004	<0.0001	0.0136	0.2379
	Location	<0.0001	<0.0001	0.0004	0.0644
	Interaction	<0.0001	<0.0001	0.0007	0.2379

^a–c^ Different superscript letters indicated a significant difference between the different aging methods (*p* < 0.05). ^1^ Below detection limit. Different aging treatments: wet-aging (WA), conventional dry-aging (DA) and UV-light dry-aging (UDA). SEM: standard error of means.

**Table 4 foods-10-02503-t004:** Consumer panelist perceptions on dry-aging and willingness to pay (*n* = 120).

End Survey Questions	Response Options	Frequency (%)
Have you ever eaten dry-aged products?	Yes	52.5
	No	14.2
	Not Sure	33.3
If you have eaten dry-aged product, where did you purchase the product from?	Local butcher store	19.1
	Local retail/supermarket	23.8
	Restaurant	39.7
	Other	17.5
If you answered “Other” in the previous question, where did you get the product from?	Personally made	45.5
	Research panels/projects	36.4
	School events	18.2
Is aging a positive or negative term?	Positive	85.8
	Negative	14.2
Do you think dry-aged product is safe?	Safer	10.8
	Less Safe	2.5
	Same as other product	65.8
	Not sure	20.8
Would you be willing to pay $1.00 more per 1 lb. of dry-aged pork?	Yes	55.8
	No	44.2

**Table 5 foods-10-02503-t005:** Effect of different aging treatments on consumer sensory panel (*n* = 120) for likeness, acceptability, and perceived quality of pork loins (*M. Longissimus lumborum*) aged for 21 days.

Traits	WA	DA	UDA	SEM	*p*-Value
** *Likeness* **					
Flavor	63.79	62.15	61.03	2.43	0.6184
Tenderness	61.53	61.80	60.78	3.04	0.9621
Juiciness	66.02	65.31	67.31	2.38	0.7876
Overall	62.99	62.72	63.89	2.60	0.9315
** *Acceptability* **					
Tenderness Acceptability	85.26	87.52	88.14	3.59	0.7950
Juiciness Acceptability	76.29	77.70	79.14	4.52	0.8762
Flavor Acceptability	86.26	82.33	84.14	3.77	0.7152
Overall Acceptability	82.14	83.62	85.09	3.62	0.8366
** *Perceived Quality* **					
Unsatisfactory Quality	13.82	15.47	13.82	3.48	0.9146
Everyday Quality	48.22	50.85	48.22	4.90	0.8981
Better Than Everyday Quality	25.25	30.68	30.39	4.36	0.5861
Premium Quality	8.00 ^a^	1.23 ^b^	4.49 ^a,b^	3.05	0.0416

^a,b^ Different superscript letters indicated a significant difference between the different aging methods (*p* < 0.05). Different aging treatments: wet-aging (WA), conventional dry-aging (DA) and UV-light dry-aging (UDA). SEM: standard error of means.

**Table 6 foods-10-02503-t006:** Effect of different aging treatments on metabolomics profile of pork loins (*M*. *Longissimus lumborum*) aged for 21 days (*p*-value < 0.05 and FDR < 0.05).

Mass	RT	HMDB ID	Name	Highest Abundance	WA	DA	UDA
** *Protein-derived* **							
155.0350	0.69	HMDB0000177	l-Histidine	DA/UDA	4.43 ^b^	4.85 ^a^	4.73 ^a^
226.0959	5.76	HMDB0001904	3-Nitrotyrosine	DA/UDA	5.20 ^b^	5.37 ^a^	5.32 ^a^
157.1467	19.41	HMDB0000459	3-Methylcrotonylglycine	UDA/DA	5.10 ^b^	5.17 ^a^	5.18 ^a^
165.1162	4.41	HMDB0000159	l-Phenylalanine	UDA	6.20 ^b^	6.21 ^b^	6.31 ^a^
129.0425	0.88	HMDB0000267	Pyroglutamic acid	WA	5.73 ^a^	5.63 ^b^	5.61 ^b^
145.1101	0.79	HMDB0003464	4-Guanidinobutanoic acid	WA	6.18 ^a^	6.00 ^b^	6.01 ^b^
181.1018	2.31	HMDB0000158	l-Tyrosine	WA/DA	6.07 ^a^	6.02 ^a^	5.96 ^b^
** *Carbohydrate-derived* **							
260.1372	4.86	HMDB0000124	Fructose 6-phosphate	DA/UDA	5.11 ^b^	5.21 ^a^	5.18 ^a^
164.0475	2.31	HMDB0000174	l-Fucose	WA	7.34 ^a^	7.28 ^b^	7.23 ^c^
** *Nucleotide-derived* **							
128.1316	0.52	HMDB0000079	Dihydrothymine	DA/UDA	6.41 ^b^	6.46 ^a^	6.45 ^a^
242.1268	5.40	HMDB0000273	Thymidine	DA/UDA	5.68 ^b^	5.85 ^a^	5.83 ^a^
329.1949	7.60	HMDB0000058	Cyclic AMP	DA/UDA	5.49 ^b^	5.68 ^a^	5.66 ^a^
348.0591	1.70	HMDB0000175	Inosine monophosphate	DA/UDA	5.23 ^b^	6.17 ^a^	6.06 ^a^
136.0387	4.12	HMDB0000157	Hypoxanthine	UDA/DA	7.45 ^b^	7.53 ^a^	7.58 ^a^
243.1835	18.87	HMDB0000089	Cytidine	UDA/DA	5.55 ^b^	5.61 ^a^	5.62 ^a^
79.0424	1.58	HMDB0000926	Pyridine	WA	6.00 ^a^	5.78 ^b^	5.78 ^b^
135.0665	1.65	HMDB0000034	Adenine	WA	5.41 ^a^	5.20 ^b^	5.15 ^b^
252.1108	4.17	HMDB0000071	Deoxyinosine	WA	5.17 ^a^	4.74 ^b^	4.63 ^b^
** *Others* **							
85.0892	4.00	HMDB0002039	2-Pyrrolidinone	DA/UDA	6.11 ^b^	6.24 ^a^	6.20 ^a^
212.0800	1.05	HMDB0014814	Benzyl benzoate	DA	4.88 ^b^	5.00 ^a^	4.94 ^a,b^
132.0247	1.28	HMDB0001844	Methylsuccinic acid	WA	7.14 ^a^	7.05 ^b^	7.02 ^b^
84.0213	0.68	HMDB0001184	Methyl propenyl ketone	WA	4.87 ^a^	4.72 ^b^	4.75 ^b^
110.9992	0.51	HMDB0002434	Hydroquinone	WA/DA	6.16 ^a^	6.15 ^a^	6.13 ^b^
122.0371	2.31	HMDB0001406	Niacinamide (vitb3)	WA/DA	6.17 ^a^	6.12 ^a^	6.07 ^b^
174.1133	0.63	HMDB0003070	Shikimic acid	WA	5.42 ^a^	5.31 ^b^	5.26 ^b^
226.1075	0.65	HMDB0000245	Porphobilinogen	WA	6.92 ^a^	6.88 ^b^	6.85 ^b^
271.1644	1.05	HMDB0003263	Pelargonidin	WA/DA	5.15 ^a^	5.12 ^a^	4.92 ^b^

^a–^^c^ Different superscript letters indicated a significant difference between the different aging methods (*p* < 0.05). Different aging treatments: wet-aging (WA), conventional dry-aging (DA), and UV-light dry-aging (UDA).

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
