# Peer review of "Effect of Dry-Aging on Quality and Palatability Attributes and Flavor-Related Metabolites of Pork Loins"

_foods, 2021, doi:10.3390/foods10102503_

Round 1
Reviewer 1 Report
An interesting descriptive article, however, sensorial section can be written in a more understandable way.
- Abstract
Missing information about TBARs results and the display stability
L15: “2ºC” must be written as 2 ºC.
L20: ”…able to differentiate any sensory differences…” Redundant
L22-24: “These findings….microbiological attributes”. The statement is not supported by the consumers panel evaluation. Should be rewritten or better explained.
- The introduction
Mixing information about UV-Light dry-aging. Have someone did before? Advantages or disadvantages? Why is it decided to do in this study?
L51 and L69: “pre-cursors” should be written as precursors
L67: “…attributes”. Needed the references of the limited studies.
- Material and methods
Section 2.1. “Sample collecting, preparation and processing”.
L76: “At 7 days post-mortem…” How were animals stored during those 7 days?
L77: “…10 market weigh pork…” Were the pork raised in the same conditions. Age of slaughter? and slaughter weight? In general add, information about the animals.
L77. “…were obtained from one side of…” Left or right side? It is also recommended to indicate the vertebrae range from which samples were obtained.
L79: “…initial microbiological…”. Was it done similarly as indicated in section 2.8. If yes, indicate. If not, explain.
L80: “…three equal sections…”. Indicate the approximate size of the sections.
L84-85: “21 days of aging, 2ºC, 70% relative humidity…” Why the aging was performed under such conditions? Do you have a reference to support this?
L84: “…70% relative humidity” In the abstract it is indicated 65% relative humidity; which one is correct?
Section 2.4.” Water-holding capacity”.
L114-115. “…using the Honikel method, described by Kim…” The original reference of the method must be cited and if needed summarize it on the manuscript.
L115: “Honikel et al (2017)[18]”. The year must be removed. The same happens in many references on the paper.
L117-118. “Samples were blotted cry using paper towels before measuring the final weight…” It is already explained in lines 112-113.
L127: “…to an internal temperature of 71ºC…” Were the samples turned when half of this temperature was reached?. It is a coomon practice.
Section 2.5. ”Warner-Bratzler shear force measurement”.
It is missing the assay parameters such as cross-head speed, load cell….
Section 2.6. ”Display color stability”.
Include a short introduction about why this is done in the whole experimental design.
Section: 2.7. “Lipid oxidation”.
L153: “… of the samples was measured through...” Were the samples grinded for running the assay? Indicate
L154: “… described by Setyabrata…”. Replace by the original reference: “Buege JA, Aust SD. Microsomal lipid peroxidation. Methods Enzymol. 1978; 52:302-10. doi: 10.1016/s0076-6879(78)52032-6. PMID: 672633.”
Briefly explain the method. Was a calibration curve prepared? In which range? Steps of the method?
Section: 2.8. “Microbial analysis”.
Maybe, it is interesting to remain the parts of the samples from which to already taken samples for the microbial analysis. Up until now it is not clear.
L161: “…Setyabrata et al.” Reference is not appropriate (cite from conference abstract). Include a microbiological standard.
L164: “…, serially diluted and…” Which dilution factor was used for the serial dilutions? How many serial dilutions were done?
Section: 2.9. “Consumer sensory analysis”.
L180-181: “The consumers sensory…”. Indicate information about age range.
L192: Describe the hedonic test: like-dislike.
Section: 2.10. “Consumer sensory analysis”.
L200: “A total of 5 samples were randomly” Why only 5 samples were chosen and not all of them were tested if available?
L203-204. “…an equal amount of chloroform and methanol were added… ” Which volumes were added?
L206: “…water was added…”. Quantity?
L205: “…extracted using a Precellys 24 tissue..” Define extraction conditions?
L231. Was an internal standard used?
L249: It is missing the method for factor extraction from PCA; cluster methos for HCA and post-hoc test.
- Results and discussion
Section 3.1.
L253.”…saleable yield…” Unify nomenclature in the test and in the table.
L263: “Our current results were in agreement with Berger et al, where…” Show the values obtained by Berger et al indicating the animal, carcass part tested, and the aging procedure done by them to allow comparison.
Table 1 and table 2: Significant digit must be revised mainly in the SEM values.
L280-283: On the statement are missing the exact data obtained by the authors you refer to.
L295-296: Revise the whole paragraph. Comparison with other works is adequate but should be properly done:
Hwang and Hong 2020 treated to HPP the meat prior to aging.; this should be indicated.
The results of Juarez et al are not the same as those reported on the paper. Indeed, the aging time was 2,7 and 14 days.
Berger et al is defined as reference 11. It is not correct.
Kim et al is defined as reference 23. It is not correct.
L318-320 and L324-326: show data from the other authors to allow reader to have them in a fast way.
L338: When you discuss about the term “dicoloration” which of the colorimetry parameters are you focusing on? Explain.
Figure 1: figure legends (a)CIE l*, (b)CIE a*…. are redundant. They are reflected in the y axis.
L360: “While limited, the presence 359of the ROS could then accelerate further oxidation following the dry-aging process.” Add a reference supporting this.
L364-365: “Those authors…the display”. It is commented the results obtained by the authors for color but in the paragraph you are explaining results for the TBARs.
L374: Microbial analysis. In general, it is not clear along the manuscript from which part of the loins you take the samples for microbial analysis and the procedure for it.
L411: “The consumer panelists responded that they mainly consumed pork 1-5 times/week (87.5%)” This information should be deeper explained. Which percentages for 1,2,3,4 and 5 times/week.
Table 3: “Table 3. Effect of different aging treatments on total aerobic bacteria (APC)”. “Total” with capital letter.
Table 4: Should be supplementary material.
Table4: Check the percentages for each question. For example, the gender percentage do not reach the 100%.
L428-429: “It is of interest to note that the majority of the panelists rated flavor as the most im-428portant palatability attribute when consuming pork products compared to tenderness and juiciness”. It is not clear this result from the results that are showed in the tables or in the test. This should be correlated with the data.
L459-460. “This observation could potentially be attributed to the unfamiliarity of consumers with the dry-aged pork”. This statement can not be reached from the results showed in the manuscript. The preferences and acceptability have been practically the same for the 3 products. The authors' interpretation is not correct (that the panel is not "familiar"). The interpretation is that the panel does not care what product it tests.
Reviewer 2 Report
This manuscript presents important research on a topic of high relevance to the meat science community and meat industry. It provides an important analysis of the role which dry ageing plays in optimisation of pork meat quality and in particular presents highly innovative aspects in the metabolomic approach taken.
The introduction needs to include some information on UV treatment and the hypothesis for inclusion in this study.
There is no mention of marbling/intramuscular fat in the manuscript. Is there any data to include on this? It is important to provide some comment on this component and any potential influence (or not) it might have on the results.
Materials and Methods: Please provide clarification on the preparation of the section used for the experiment. Line 76 suggests it is just the LL but later, e.g. ‘from loin eye portion’ Line 79, text suggests that the section contains more than just the LL.
Provide a comment to clarify if all samples for metabolomics etc were taken internally i.e. not close to surface
Was one sample taken from each side for microbiology, or was a sample taken after the sectioning? Please clarify.
Provide equipment details for dry agers and UV treatment. Provide details of what type of surface sections were placed on e.g. rack, solid shelf.
Line 87: ‘Sections were rotated’ Does this mean the location in the dry ager was rotated or the individual sections were rotated.
Line94: Were chops for tenderness analysis stored at -80C or should this read -20C? If -80C was used please provide an explanation why this temperature was chosen when -20C would be more typical for this type of study. What influence might this have had on e.g. thaw loss etc.
Line 114: were efforts made to keep geometry of meat pieces similar?
WBSF method – provide crosshead speed.
Results and Discussion:
Line 282: change ‘microbial’ to ‘microbes’?
Line 511…: the pH values observed in this study were ~5.6. The authors discuss the fact that this is in the lower range. However this would seem to be more of a typical/normal ultimate pH rather than being lower. Please review and reword this paragraph to take this on board.
circa Line 533: it is not clear if the meat used for the metabolomics was internal in the LL muscle and if so how deep it might be. UV would be expected to have a surface level effect or a least not penetrate deep in the muscle. It is important that the authors discuss this in more detail. Would they anticipate a UV treated effect on the surface only, and would they anticipate the effect of this treatment would penetrate far into the muscle e.g. indirectly via ROS migration.
Conclusions: include comment on UV treatment.
